# *Ab initio* calculation on Herbertsmithite: exchange interactions including extra-plane magnetic impurities, Dzyaloshinskii-Moriya and anisotropic coupling

Authors: Flaurent Heully-Alary,[1] Nadia Ben Amor, [1] Nicolas Suaud, [1] Laura Messio,[2] Coen de Graaf[3,4] and Nathalie Guihéry[1] *

Affiliations:

(1) Laboratoire de Chimie et Physique Quantiques, UMR5626, CNRS, Université de Toulouse, 118 route de Narbonne, 31062 Toulouse cedex 4, France.
(2) Laboratoire de Physique Théorique de la Matière Condensée (LPTMC), CNRS (UMR 7600), Sorbonne Université, 4 place Jussieu F-75005 Paris, France.
(3) Department of Physical and Inorganic Chemistry, Universitat Rovira i Virgili, Marcelli Domíngo 1, 43007 Tarragona, Spain.
(4) ICREA, Pg. Lluís Companys 23, 08010 Barcelona, Spain.

*nathalie.guihery@irsamc.ups-tlse.fr

Abstract: A detailed ab initio evaluation of the isotropic and anisotropic exchange interactions of Herbertsmithite is presented. This compound crystallizes in a so-called Kagome lattice and the $S=\frac{1}{2}$ spin structure is not fully resolved despite numerous experimental and theoretical studies. The present study not only focusses on the leading in-plane nearest neighbor isotropic interactions, but also other less well studied interactions such as the anisotropic exchange and the Dzyaloshinskii-Moriya (DM) interactions. The anisotropic exchange is very weak, but the DM interactions are sizeable with a strong in-plane component, typically obviated in model studies. Moreover, it is shown that the extra-plane magnetic impurities have a non-negligible interaction with the regular in-plane magnetic sites. Combined with an estimated occurrence of these magnetic impurities of ~15%, the present results indicate that two-dimensional magnetic models only describe part of the physics.

## I. Introduction

The field of frustrated magnetism, in its ongoing search for new layered Mott insulators, remains fascinated by a now quite old one: Herbertsmithite $ZnCu_3(OH)_6Cl_2$ [1,2]. In a first approximation, it is described by one of the most puzzling, yet extremely simple Hamiltonians: the Heisenberg model, which consists in magnetic exchange couplings of S = 1/2 spins on neighboring sites of a Kagomé lattice.

Already at the classical level, this model remains unordered down to zero temperature (classical spin liquid), with an extensive ground state degeneracy due to a flat band of excitations [3]. This behavior is expected to persist in the quantum case. Thermal and quantum fluctuations have been extensively studied, both at the classical [4–8] and at the quantum level [9,10], as they are expected to favor certain states, such as the coplanar ones, caused by what is known as the order by disorder effect.

At the quantum level, the exact nature of the S = ½ Kagomé antiferromagnet is today still an open question, which has stimulated many theoretical and numerical studies (high temperature series expansions [11,12], exact diagonalizations [11,13], tensor network methods [14–16], mean-field approaches [17], variational studies [18,19]...). After a time when the balance tipped towards a gapped $\mathbb{Z}_2$ spin liquid, the current trend is towards an algebraic spin liquid ground state, with no gap. The $U(1)$ Dirac spin liquid is one such serious candidate.

Back to Herbertsmithite, the excitement of having a compound that realizes such an interesting theoretical model has led to increasingly advanced synthesis methods (high quality crystals [20]) and measurements (specific heat under high magnetic fields [21], NMR magnetic susceptibility [22,23], thermal conductivity, magnetic structure factors via neutron scattering [24]). The precision achieved today highlights the inevitable deviations of the compound from the ideal theoretical model.

Many different deviations can occur, and their effect is inevitably strong due to the high density of low energy states. For example, further neighbor interactions lift the classical degeneracy: second neighbors favor the $\mathbf{q} = 0$ or $\sqrt{3} \times \sqrt{3}$ long-range order, while interactions beyond second neighbors open a wide range of exotic classical orders, including chiral ones [25,26], some of which were eventually realized in other Kagomé compounds [27,28].

The most widely discussed deviation is the Dzyaloshinskii-Moriya (DM) interaction, experimentally detected by ESR [29–31]. Its origin is relativistic as well as that of the symmetric tensor of anisotropy of exchange $D_{ij}$. Once included, the anisotropic spin Hamiltonian reads:

$$H = \sum_{<i,j>} \left( J_1 \mathbf{S}_i . \mathbf{S}_j + \mathbf{S}_i . \overline{\overline{D}}_{ij} . \mathbf{S}_j + \mathbf{d}_{ij} . \mathbf{S}_i \wedge \mathbf{S}_j \right) \qquad (1)$$

where i and j are first-neighbor magnetic sites. Theoretical investigations considering the DM have shown that magnetic order appears above $|d_{ij}^\perp / J| \gtrsim 0.1|$ for out-of-plane $d_{ij}^\perp$ vectors [17,32–35], suggesting a smaller value for Herbertsmithite. The in-plane DM component $d_{ij}^\parallel$ is expected to be even smaller and is mostly neglected in theoretical studies (it complicates the problem since the total spin along the out-of-plane direction is no longer conserved) with notable exceptions [36,37].

Another relevant perturbation is the occupation disorder [38], which has important consequences. Two types of substitutions occur in Herbertsmithite: the first are magnetic vacancies within the Kagomé plane, where $Cu^{2+}$ ions are replaced by $Zn^{2+}$, and the second is the opposite, where magnetic impurities consisting of $Cu^{2+}$ replace inter-plane $Zn^{2+}$. While the first type seems to be quite rare, the second one has an occurrence rate of about 0.15 [23,39,40]

with strong effects [41]. However, most theoretical studies are limited to the simpler problem of in-plane magnetic vacancies [42–48].

To allow more predictive and well-oriented theoretical studies, it is important to have good estimates of the parameters of the model describing Herbertsmithite. It is known that the overall scale is $J_1 \simeq 180K$. DFT calculations have been performed [49] to evaluate eight different exchange couplings, including inter-plane ones, but DM was not evaluated *ab initio* up to now. However, *ab initio* calculations combined with the effective Hamiltonian theory allow to extract all interactions of the model Hamiltonian of equation (1) [50–59].

In this article, we tackle the first *ab initio* evaluation of the anisotropy tensor and of the DM vector for Herbertsmithite, as well as the evaluation of the exchange between Kagomé $Cu^{2+}$ sites and $Cu^{2+}$ inter plane impurities.

## II.     Theory

The approach used here consists in extracting the local effective interactions of the model Hamiltonian of equation (1) extended to magnetic exchange interactions between non-first-neighbors from calculations performed on embedded clusters. The embedded cluster procedure enables correlated calculations, including relativistic treatment if required, to be carried out on small fragments immersed in a realistic representation of the environment. Two types of calculations were carried out: i) density functional theory (DFT) calculations on clusters of different sizes and shapes to determine the main magnetic couplings between copper ions, ii) *ab initio* calculations based on wave function theory (WFT) including electron correlation effects and spin-orbit couplings to determine anisotropic interactions.

*Embedded cluster approach.*

The embedded cluster procedure takes into account the effects of the crystal environment by immersing a fragment of the material in a set of optimized point charges which accurately reproduces the Madelung field of the crystal. Total Ion Potentials (TIPs) were used to represent the immediate neighboring ions of atoms located on the edge of the explicitly treated cluster. The reliability of this procedure to estimate magnetic exchange parameters has been established in numerous previous studies, and has even led in some cases to the questioning of commonly used models, followed by the proposal of more appropriate ones [60,61]. The value of the main magnetic coupling $J_1$ (see Figure 1) obtained in this local approach was compared with that obtained in a periodic calculation (see computational information) to check the quality of the embedding. Moreover, the stability of the results can be assessed by comparing the parameters values obtained for different clusters. These verifications ensure that appropriately embedded cluster containing a small number of magnetic centers and their immediate neighbors provides a sufficiently reliable representation of the material for a correct description of the local electronic structure.

*Broken-symmetry DFT calculations for the determination of isotropic magnetic couplings.*

Calculations were performed on five clusters of different sizes and shapes which are depicted in Figure 1. The scheme of the in-plane interactions is provided in Figure 2. Broken-spin symmetry DFT (BS-DFT) solutions have been computed by imposing the +1/2 or -1/2 value of the *Ms* component of the spin moment on each copper ion. Their energy differences were

assimilated to those of the Ising Hamiltonian. Such a procedure generates different sets of equations from which the $J_1$, $J_2$, $J_3$, $J_4$ and $J_5$ magnetic couplings can be extracted. For instance, for the 13-copper, 11 BS-DFT solutions have been computed, from which 96 sets of independent equations have been generated. Among them, 35 sets calculated from the most antiferromagnetic solutions (reversing 3 or 4 spins from the fully ferromagnetic solution) have been retained as they provide consistent values for all the interactions.

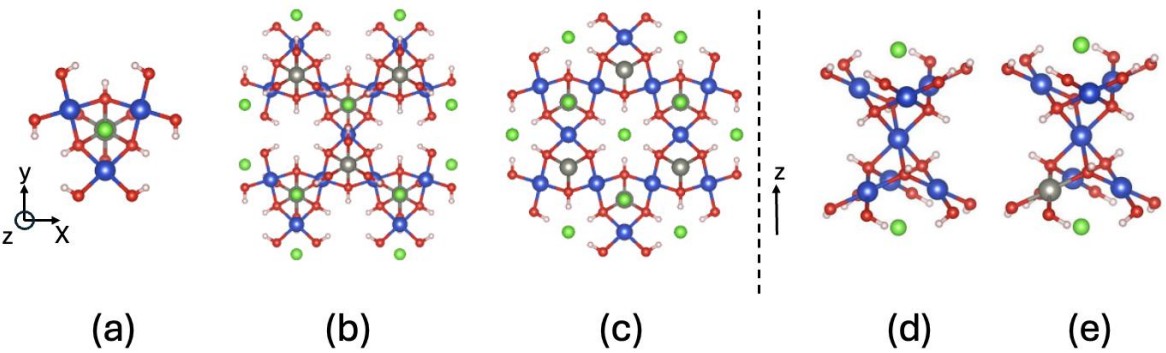

Figure 1: Clusters considered in the DFT calculations: (a) 3-copper cluster, (b) 13-copper cluster, (c) 12-copper cluster. (a), (b), (c) consider the in-plane copper only. (d) 7-copper cluster with an inter-plane $Cu^{2+}$ at the position of the $Zn^{2+}$ ion, (e) 6-copper cluster where the inter-plane $Zn^{2+}$ and an in-plane $Cu^{2+}$ ions have been interchanged. Copper in blue, oxygen in red, hydrogen in white, zinc in grey and chlorine in green.

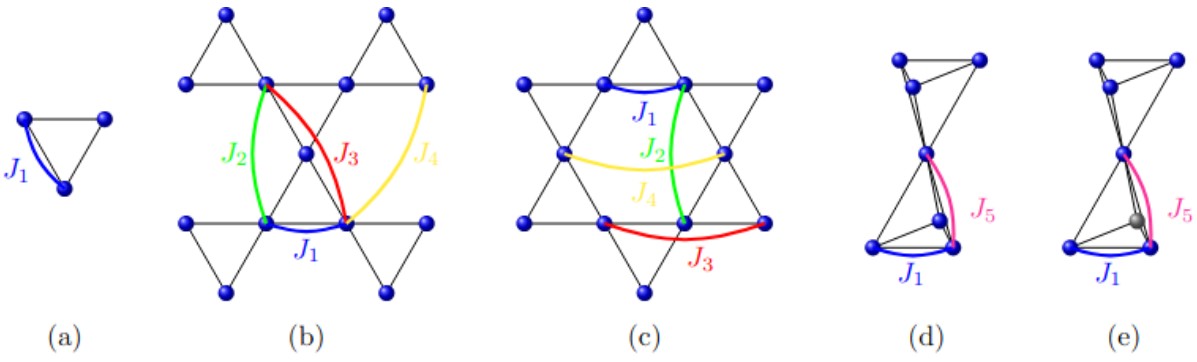

Figure 2: Scheme of the magnetic couplings for the clusters (a), (b), (c), (d) and (e).

*Wave-Function Theory calculations for the determination of anisotropic interactions.*

To extract the anisotropic interactions, we have performed relativistic correlated wave-function based *ab initio* calculations. To check the computational procedure, we first reproduced the value of $J_1$, being the leading interaction in this material. For this purpose, we considered the embedded cluster (a). We performed Complete Active Space Self Consistent Field (CASSCF) calculations to account for non-dynamic correlation effects, i.e. the wave-functions contain at least all possible distributions of the magnetic electrons in the magnetic orbitals. The active space CAS(9,6) hold 9 electrons in 6 orbitals, namely the three magnetic orbitals of the $Cu^{2+}$ ions and 3 doubly occupied orbitals located on the bridging oxygens. These last orbitals have been obtained by projecting the magnetic orbitals onto the *p* orbitals of the bridging oxygens

according to a procedure which is known to furnish good bridging orbitals for the calculation of exchange couplings [62]. Dynamic correlation effects, i.e. the correlation of all other electrons, are included through variational calculations using difference dedicated configuration interaction (DDCI) [63], which is one of the most accurate available *ab initio* methods for the calculation of exchange couplings [50] (see also computational information). To determine the two-body (two $Cu^{2+}$ magnetic centers) anisotropic interactions $\overline{\overline{D}}_{ij}$ and $\boldsymbol{d}_{ij}$, relativistic calculations have been performed on the cluster (a) where one $Cu^{2+}$ ion has been replaced by a $Zn^{2+}$. The here-used Spin-Orbit State-Interaction (SO-SI) method [64] diagonalizes the spin-orbit matrix in the basis of all $Ms$ components of the spin states $S$ calculated at the CASSCF level. All Cu-3$d$ orbitals were introduced in the active space resulting in a CAS(18,10). This enlarged active space is necessary to account for the SO couplings of the ground spin-orbit free state with all excited states of the configuration. To introduce dynamic correlation on all states, we performed multireference second-order of perturbation CASPT2 [65] calculations and used the dynamically correlated energies of the spin-orbit free states as diagonal elements of the SO matrix. To check the transferability of the anisotropic interactions, we also performed calculations on a triangular fragment involving three Cu2+ ions. To account for the coupling with all the states of the configuration, a CAS(27,15)SCF calculation containing all the $d$ electrons in all Cu-3$d$ orbitals was performed.

To extract the anisotropic exchange tensor components and the DM vector from the *ab initio* results, we used the effective Hamiltonian theory [66,67]. This theory has been specially adapted to the extraction of anisotropic interactions and has been shown to provide very reliable values of the parameters of anisotropy in mono- and bi-nuclear complexes [51,59,68–71]. It consists in defining a bijective relation between the target space, made up of the states calculated *ab initio*, and the model space that spans the model Hamiltonian. The effective Hamiltonian obeys the following relation:

$$H^{Eff\cdot}|\tilde{\psi}_i\rangle = E_i|\tilde{\psi}_i\rangle \qquad (2)$$

where $|\tilde{\psi}_i\rangle$ are the orthogonalized projections of the *ab initio* SO states $|\psi_i\rangle$ onto the model space and $E_i$ are the energies of the SO states calculated *ab initio*. The numerical matrix of the effective Hamiltonian can be calculated from its spectral decomposition:

$$H^{Eff\cdot} = \sum_i E_i |\tilde{\psi}_i\rangle\langle\tilde{\psi}_i| \qquad (3)$$

and then compared to the analytical matrix of the model Hamiltonian. The extraction is based on a term-by-term comparison of numerical matrix elements of the effective Hamiltonian and analytical elements of the model Hamiltonian of equation (1) that reads:

$$(H) = \begin{array}{c} \\ \langle T^+| \\ \langle T^0| \\ \langle T^-| \\ \langle S| \end{array} \begin{pmatrix} \frac{J_1}{4}+\frac{D_{zz}}{4} & \frac{D_{xz}-iD_{yz}}{2\sqrt{2}} & \frac{(D_{xx}-D_{yy}-2iD_{xy})}{4} & \frac{d_y+id_x}{2\sqrt{2}} \\ \frac{D_{xz}+iD_{yz}}{2\sqrt{2}} & \frac{J_1}{4}-\frac{D_{zz}}{4}+\frac{(D_{xx}+D_{yy})}{4} & -\frac{D_{xz}-iD_{yz}}{2\sqrt{2}} & -\frac{id_z}{2} \\ \frac{(D_{xx}-D_{yy}+2iD_{xy})}{4} & -\frac{D_{xz}+iD_{yz}}{2\sqrt{2}} & \frac{J_1}{4}+\frac{D_{zz}}{4} & \frac{d_y-id_x}{2\sqrt{2}} \\ \frac{d_y-id_x}{2\sqrt{2}} & \frac{id_z}{2} & \frac{d_y+id_x}{2\sqrt{2}} & -\frac{3J_1}{4}-\frac{D_{zz}}{4}-\frac{(D_{xx}+D_{yy})}{4} \end{pmatrix} \qquad (4)$$

with column headers $|T^+\rangle \quad |T^0\rangle \quad |T^-\rangle \quad |S\rangle$

where $T^+, T^0, T^-$ are the $Ms$=1, Ms=0, $Ms$=-1 components of the lowest spin-orbit free triplet state $T$ respectively and $S$ is the lowest spin-orbit free singlet state; $J_1$ is the isotropic magnetic exchange ; $D_{xy}, D_{xz}$ , $etc.$ are the components of the symmetric $\overline{\overline{D}}_{ij}$ tensor of exchange anisotropy and $d_x$, $d_y$ and $d_z$ the x, y and z components of the DM vector. The DM vector components are directly extracted from the numerical matrix elements between the singlet and the three $Ms$ components of the triplet. The symmetric exchange tensor is determined from the numerical interactions between the triplet components. After diagonalizing the resulting matrix of the tensor and imposing a zero trace, the symmetric exchange reduces to two terms only: the axial and rhombic parameters $D$ and $E$.

For the extraction on a triangular fragment involving three $Cu^{2+}$ ions, the numerical effective Hamiltonian matrix was built in the basis of eight spin-orbit-free functions: the four Ms=-3/2, -1/2, 1/2, 3/2 components of the quadruplet and the two Ms=-1/2, +1/2 components of the two doublet states. It allows to determine the three symmetric and antisymmetric (DM) tensors of the three couples of $Cu^{2+}$ ions.

*Computational information.*

The geometrical structure for all heteroatoms has been taken from the X-Ray study published in reference [72]. Periodic Density Functional Theory (DFT) calculations have been performed with the quantum espresso [73] code and the PBE [74] functional to optimize the hydrogens positions.

Concerning the embedding, effective Core Potential (ECP) (for DFT calculations) and *ab initio* model potentials (AIMP) (for WFT) have been introduced to represent the ions close to the atoms of the cluster. We checked the accuracy of the embedding by comparing the B3LYP results for the $J_1$ interaction in the embedded cluster and in periodic calculations performed with the Crystal [75] code also using the B3LYP [75,76] functional. The $J_1$ values are in perfect agreement: periodic = 240 K and embedded cluster = 239 K. As we will see below, these B3LYP values slightly overestimate the coupling but demonstrate the adequacy of the material model adopted in this study. Further details on the embedded cluster method can be found in reference [78].

For the DFT cluster calculations, we used the ORCA code [79] with the def2-TZVP basis set [80] for all atoms and the $\omega$B97X-D3 functional [81] which has been shown to provide very good values of magnetic couplings [82].

The WFT calculations were carried out with the MOLCAS [83] and CASDI [84] codes. We used extended basis sets of ANO-RCC type [85,86] (6s5p3d2f for Cu and Zn, 4s3p2d for O, 4s3p1d for Cl and 2s1p for H). DDCI1 calculations were performed on the top of the active space CAS(9,6) enlarged with the bridging $p$ orbitals of the oxygen, as recommended in references [87].

Finally, CAS(2,2)SCF calculations were performed on a bi-nuclear fragment to get a picture of the magnetic orbitals well-localized on just two copper ions. Their orientation and decomposition on metal and ligand are identical to the naked eye, to those obtained for larger fragments, or larger active spaces. CAS(4,4)SCF calculations were performed on a tetra-nuclear fragment involving three in-plane and one inter-plane copper ions for the same reason.

## III.    Results and discussion

*Isotropic couplings.*

Table 1 reports the isotropic exchange couplings obtained for the six clusters of Figure 1. One should note the good transferability of the interaction values when increasing the size or changing the shape of the clusters (comparison of clusters (a), (b) and (c)). This transferability is also observed when substituting a $Zn^{2+}$ by a $Cu^{2+}$ or by exchanging a $Cu^{2+}$ and a $Zn^{2+}$ (comparison of clusters (d) and (e)). The $J_1$ interaction is in good agreement with values published in the literature and the antiferromagnetic $J_2$ and ferromagnetic $J_3$ and $J_4$ interactions are very weak, in line with the long distances between the $Cu^{2+}$ ions involved in these interactions. Note that the $J_4$ coupling is particularly weak despite the distance between the copper ions being identical to that of $J_3$. This is because the interactions between the copper ions involved in the exchange pass through one copper (and its surrounding oxygens) for $J_2$ and $J_3$, whereas they pass through two copper ions in the case of $J_4$.

The most important result of this series of cluster calculations is the high ferromagnetic value of $J_5$. Note the small distance between the in-plane and inter-plane ions, indicating that the copper ions involved in this interaction are in fact closer than those of the $J_1$ exchange. To obtain insight into the ferromagnetic character of the interplane couplings, we have plotted in Figure 3 one magnetic orbital derived from a CAS(4,4)SCF calculation on a tetra-nuclear embedded fragment (obtained by removing the three uppermost copper ions of cluster d). The nature of the coupling is roughly determined by three ingredients: (i) orbital overlap increasing the kinetic exchange and hence the antiferromagnetic character of the interaction; (ii) the direct exchange, a purely ferromagnetic interaction; and (iii) the spin polarization, that may favor ferro- or antiferromagnetism depending on the system [50]. The local *d*-magnetic orbital of the in-plane copper ions exhibits only a very small overlap with that of the inter-plane one, excluding a large kinetic exchange effect. The direct exchange integral is generally quite small between distant atoms, and therefore, large value of $J_5$ must be attributed to spin polarization. The appearance of such a large interaction calls into question the validity of considering this material as two-dimensional.

| $J_i(K)$ | (a) | (b) | (c) | (d) | (e) | $d_{CuCu}(\text{Å})$ |
|---|---|---|---|---|---|---|
| $J_1$ | 178.0 | 191.2 | 181.0 | 184.1 | 182.7 | 3.42 |
| $J_2$ | - | 0.5 | 0.4 | - | - | 5.91 |
| $J_3$ | - | -1.1 | -1.0 | - | - | 6.83 |
| $J_4$ | - | -0.1 | -0.2 | - | - | 6.83 |
| $J_5$ | - | - | - | -83.4 | -82.0 | 3.07 |

Table 1: $\omega$B97X-D3/def2-TZVP values of the interactions computed for the 5 clusters and distances between the $Cu^{2+}$ involved in the interaction.

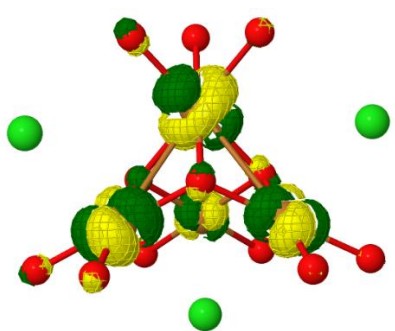

Figure 3: CAS(4,4)SCF magnetic orbital calculated for a tetra-nuclear fragment constituted of three in-plane (below) and an inter-plane (above) copper ions.

*Anisotropic interactions.*

Before extracting the anisotropic exchange interactions, we first calculate the isotropic interaction $J_1$ using cluster (a). The interaction of 181.3K is in perfect agreement with the one obtained with the DFT method for the same cluster, confirming the reliability of the $\omega$B97X-D3 functional. The antiferromagnetic nature of this magnetic exchange is due to the super-exchange mechanism occurring through the bridging oxygen. Figure 4 illustrates the contribution of the oxygen *2p* orbital to the essentially *3d* magnetic orbitals of the copper ions.

All anisotropic parameters are reported in Table 2. One may first note the good transferability of the parameters extracted from either bi-nuclear or tri-nuclear calculations. The anisotropic exchange values of *D* and *E* are very small and will have only very little impact on the magnetic properties of the system. The DM vector components, on the contrary, are non-negligible. Note that the in-plane (XY) component $|d_{ij}^{\parallel}|$ of the DM vector is larger than the out-of-plane $|d_{ij}^{\perp}|$ (along Z) one. This *a priori* surprising result can, however, be rationalized. Indeed, as demonstrated analytically in reference [52], the physical origin of the DM vector is the hybridization of the metal's Cartesian *d*-orbitals. The mixing of the *d* orbitals conditions the nature of this interaction. The analytical derivation presented in Ref [52]showed that the that i) a mixing between $d_{x^2-y^2}$ and $d_{xy}$ or between $d_{xz}$ and $d_{yz}$ generates a $d_z$ component, ii) a mixing between $d_{x^2-y^2}$ and $d_{xz}$ or between $d_{xy}$ and $d_{yz}$ generates a $d_y$ component and iii) a mixing between $d_{x^2-y^2}$ and $d_{yz}$ or between $d_{xy}$ and $d_{xz}$ generates a $d_x$ component. The contribution of the out-of-plane $d_{xz}$ and $d_{yz}$ Cartesian orbitals in the magnetic orbitals is clearly evidenced in Figure 4 and can be appreciated by their coefficients in the magnetic orbitals provided in the caption of Figure 4. This rationalizes the strong in-plane component of DM vector. Its orientation is depicted in Figure 5, where we may note that its out-of-plane component $|d_{ij}^{\perp}|$ oscillates between below and above the (XY) plane of copper ions from one triangle to the next, following the alternating positions above and below of the oxygens.

| Fragment | $\|d_{ij}^{\parallel}\|$ | $\|d_{ij}^{\perp}\|$ | $\|d\|$ | $D$ | $E$ |
|---|---|---|---|---|---|
| Bi-nuclear | 4.73 | 1.70 | 5.03 | -0.46 | 0.13 |
| Tri-nuclear | 4.73 | 1.78 | 5.05 | -0.51 | 0.16 |

Table 2: In-plane $\|d_{ij}^{\parallel}\|$ and out-of-plane $\|d_{ij}^{\perp}\|$ components of the DM vector and axial $D$ and rhombic $E$ parameters of the anisotropic exchange in kelvin extracted from calculations performed on bi-nuclear and tri-nuclear fragments.

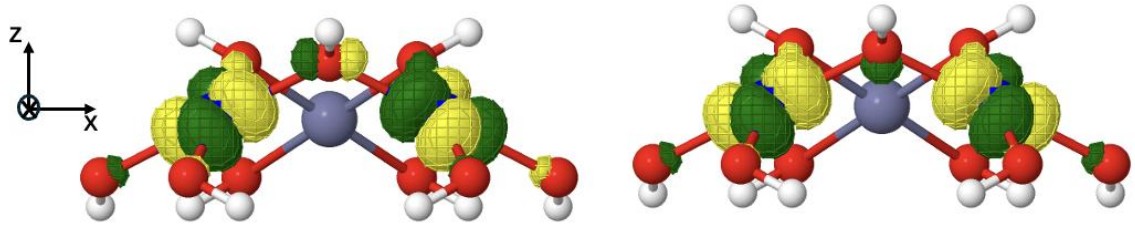

Figure 4. CAS(2,2)SCF magnetic orbital calculated on a fragment involving two in-plane copper ions. The expression of one of these magnetic orbitals on the Cartesian d orbitals of one copper ion is: $0.4889\, d_{x^2-y^2}$ $+ 0.2120\, d_{xz}$ $-0.2841\, d_{xy}$ $-0.3666\, d_{yz}$+ small coefficients on the ligands orbitals.

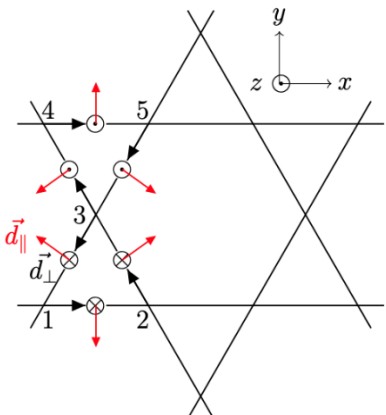

Figure 5. Picture of the DM vectors on two neighbor triangles of the lattice. The black arrows between the magnetic centers indicates the order of the first and second spins in the vector products of the DM interactions. The out-of-plane components alternate from one triangle to its neighbors and point in the opposite direction to the position of the oxygens. The angle between the in-plane components of the vectors within the triangle is 120°. The X, Y, Z components (in kelvin) of the three vectors extracted from the tri-nuclear calculations are $d_{12}$=(0, -4.73, -1.78), $d_{23}$=(4.10, 2.36, -1.78), $d_{31}$=(-4.10, 2.36, -1.78).

Summary and conclusions

The embedded cluster model is used to evaluate isotropic and anisotropic exchange interactions in Herbertsmithite, characterized by S=½ spins located on a Kagome lattice. The leading magnetic interaction is the in-plane nearest neighbor isotropic exchange. This interaction is antiferromagnetic in nature and its strength, parametrized by $J_1$, is generally considered to be ~180 K in a Hamiltonian $H = \sum_{<i,j>} (J_1 \boldsymbol{S_i}.\boldsymbol{S_j})$ i and j being first-neighbour Heisenberg model. Both the DFT and WFT values are in close agreement with this experimental result, validating the material model and the electronic structure methods used. This is in line with many previous studies of exchange interactions in similar compounds, using the same computational techniques.

Second-neighbor in-plane interactions (and beyond) are all very small and there is no need to include these interactions in model studies. Triggered by the experimental evidence for a substantial number of magnetic impurities located between the S=½ Kagome planes, we have also calculated the isotropic exchange between a regular in-plane $Cu^{2+}$ center and a copper ion replacing an interplane $Zn^{2+}$. This interaction turns out to be ferromagnetic and far from being negligible, ~-80 K. The commonly used models to rationalize the experimental observations only consider in-plane interactions, but this out-of-plane interaction questions the validity of a purely bidimensional model.

Concerning the anisotropic interactions, the results indicate that the symmetric anisotropic exchange interaction does not contribute in a significant manner to the low-energy spectrum of this material. With values smaller than 1 K for the axial and rhombic anisotropic exchange parameters, it is not expected that this interaction plays a role in the magnetic properties. This is not the case for the Dzyaloshinskii-Moriya vector (DM, or antisymmetric tensor of anisotropy). The values extracted from the combination of *ab initio* WFT calculations and effective Hamiltonian theory are on the order of several Kelvins with the in-plane component significantly larger than the out-of-plane component. The analysis of the magnetic orbitals shows that there is a sizeable mixing of the $d_{xy}$ and $d_{x2-y2}$ in plane Cartesian orbitals with the $d_{xz}$ and $d_{yz}$ one in the magnetic orbitals, caused by the out-of-plane position of the bridging oxygen ions. This mixing was shown in previous studies to originate in plane DM interactions. The three DM pseudo-vectors of the triangles formed by the $Cu_3$-O units point all in the direction opposite to the oxygen positions and result in a small net out-of-plane interaction pointing alternatively up and down for neighboring triangles.

The results derived here for Herbertsmithite are valid for a substitution rate of the Zn by Cu up to 0.66, through the Zn-paratacamite family. But for higher substitution, a structural transition has been reported [88], and the fully substituted compounds, so-called clinoatacamite, is expected to have different exchanges. A further work will be devoted to the study of this material.
Another extension of this work could evaluate how the DM interactions in the triangles of the Kagome lattice are affected by the presence of a S=1⁄2 ion completing the coordination of the bridging oxygen.

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
