# Peer review of "Is Herbertsmithite far from an ideal antiferromagnet? Ab-initio answer including in-plane Dzyaloshinskii-Moriya interactions and coupling with extra-plane impurities"

_SciPost Physics Core_

## Round 2 · Referee Report · Anonymous (Referee 1) · 2025-8-13

Strengths
- There are many experimental indications of Cu/Zn intersite mixing and/or impurity Cu between the kagome planes in herbertsmithite, yet their impact on the physics remains poorly studied, particularly from the first-principles perspective.
- Likewise, the DM anisotropy is known to influence the physics of the kagome Heisenberg model, yet no detailed information about it is known for herbertsmithite.
Weaknesses
- Possible local structural distortions are not properly considered.
- The method for estimation of the DM exchange couplings lacks proper benchmarking for undoped Cu2+ oxides.
Report
The sizable ferromagnetic exchange between the impurity spins and kagome planes exceeding -80 K is a new interesting result that indeed questions the quasi-2D nature of herbertsmithite. However, it is difficult to reconcile this result with the Weiss temperature of -5.2 K extracted from ESR data (PRL 118, 017202 [2017]). Also averaged values obtained using DFT on locally optimized structures (Tab. 7.2 in https://nbn-resolving.org/urn:nbn:de:bsz:14-qucosa-91976) are significantly smaller (between -10 and -20 K). In the latter case, a structural relaxation has been performed to simulate a Jahn-Teller distortion of an octahedron under Cu substitution. Fingerprints of a such distortion are also found in the experiment, e.g. in Ref. 23: "...a staggered response is certainly associated with a Jahn-Teller driven distortion leading to a displacement of the six adjacent oxygens". This makes me think that structural relaxation is a key ingredient that can not be bypassed if a magnetic Jahn-Teller-active impurity like Cu2+ occupies a regular octahedron.
While the DM interaction reasonably turns out to be the leading anisotropy, its magnitude -- less than 3% of the Heisenberg exchange -- seems to be rather low. Note that in La2CuO4 with a very small tilt angle (3-4 degrees), the DM anisotropy amounts to about 0.6 % of the Heisenberg exchange (PRB 108, 085140 [2023]). In herbertsmithite, the neighboring CuO4 squares are significantly more tilted, giving rise to a sizable overlap between x^2-y^2 and other d-orbitals, which is also stated in the manuscript, e.g. in the caption of Fig. 4. I would naively expect a more significant enhancement of the DM component in this case. I suggest the authors benchmark their approach by calculating D1 and J1 in La2CuO4, for which experimental estimates are available and which have been extensively studied by other numerical methods.
Minor points:
1) This becomes obsolete if the authors perform a structural optimization, as suggested earlier, but I still wonder why the structural input is taken from Ref. 74, which is a lab XRD study on natural samples. There are more recent and more detailed structural studies on synthetic samples, e.g. JACS 127, 13462 [2005], JACS 132, 16185 [2010], Appl. Phys. Lett. 98, 092508 [2011].
2) The naming convention for orbitals is somewhat vague. My understanding is that "Cartesian d-orbitals" are standard d-orbitals in some global Cartesian frame with the x axis along a Cu-Cu bond; "d^2-y^2", "dxz" etc. presumably refer to a local coordinate frame of CuO4. Please explain how these frames are defined.
3) Note that there exist other numerical DFT-based methods for calculations of DM interactions, such as the Green's-function methods (PRB 71, 184434 [2005]) and the energy-mapping approach (Dalton Trans. 42, 823 [2013]). Both have been applied to several Cu2+-containing oxides.
Requested changes
- Perform structural optimization of clusters with impurity Cu atoms.
- Benchmark the method used to estimate DM interactions by performing a similar analysis for a well-studied material such as La2CuO4.
Recommendation
Ask for major revision

---

## Round 2 · Referee Report · Anonymous (Referee 2) · 2025-8-28

Strengths
2 Manuscript demonstrates consistency of embedded cluster calculations with respect to cluster size
Weaknesses
2 The authors omit the local point group symmetry analysis of the system, which typically determine the form of the DM interaction up to numerical values.
Report
Sizable Dzyaloshinskii-Moriya (DM) interactions and ferromagnetic interactions with out-of-plane impurities question the overall consensus that the only relevant magnetic interaction in Herbertsmithite is nearest-neighbor isotropic exchange. Future studies should verify the stability of the spin-liquid state with respect to those interactions.
However, there are several problems with the manuscript.
1 It is not clear from the paper why authors decided to treat isotropic and anisotropic exchange separately with different methods. Within the effective Hamiltonian approach they employ to obtain DM interactions, they also obtain J1, which should probably be considered as their best estimates. But it is not reported in Table 2.
It should also be possible to estimate J5 and the corresponding possible anisotropic interactions with out-of-plane impurity using WF-based methods to get consistent picture.
2 In the DFT calculations of further-neighbor isotropic couplings using bigger clusters the authors opted for choosing a different basis set compare to WF-based part of the study, which is again reduces the coherency of the study.
3 In the text authors advocate for the DDCI method to be one of the most accurate WF-based methods to treat dynamical correlations for magnetic systems. The DDCI3 indeed shows very good performance for exchange couplings. But in this study the least accurate form (DDCI1) equivalent to MRCI-S (multireference configuration interaction with single excitations) is used, which is tend to overestimate magnetic couplings. This point should not be hidden, but rather discussed in the text.
4 Authors mention that the orientation of the DM vector (would be the fraction of d_parallel/d_perpendicular in the paper's notation) is "a priori surprising result", but in most cases this orientation is fully determined by the local point-group symmetry of the dimer. Such analysis should be a natural part of the paper.
There are other small things:
Authors say they use orbitals from small CAS(2,2)SCF and CAS(4,4)SCF calculations for illustration purposes instead of actual DFT or CASSCF calculations. Nevertheless, in the text they actually mention and analyze their shape. In this case the orbitals from the actual calculations should be employed.
In the table 2 the authors report results of the symmetric anisotropic exchange in a very rudimentary form. The D and E are not defined, moreover without the corresponding main magnetic axes the results are not complete.
Requested changes
1 Reason in the manuscript why anisotropic and isotropic interactions are obtained with WF-based and DFT-based calculations respectively.
2 Clearly present all couplings obtained from WF-based method.
3 Possibly use consistent basis sets and fuctionals though the study.
4 Clearly state that the employed DDCI1 method is equivalent to MRCI-S and is less accurate compare to the most of the cited DDCI studies.
5 Provide symmetry analysis of the two- (and possibly three-) magnetic-site problem with symmetry-motivated direction of the DM vector
6 Use orbitals form actual DFT and CASSCF/DDCI1 studies in the shapes analysis
7 Provide expressions for D and E in the Table 2 together with the associate main magnetic axes.
Recommendation
Ask for major revision

---

## Editorial Decision

unknown